# Advances in Polyethylene Terephthalate Beverage Bottle Optimization: A Mini Review

**DOI:** 10.3390/polym14163364

**Published:** 2022-08-17

**Authors:** Shangjie Ge-Zhang, Huixin Liu, Mingbo Song, Yanzhi Wang, Hong Yang, Haobo Fan, Yuyang Ding, Liqiang Mu

**Affiliations:** 1College of Science, Northeast Forestry University, Harbin 150040, China; 2College of Forestry, Northeast Forestry University, Harbin 150040, China; 3College of Clinical Medicine, Harbin Medical University, Harbin 150086, China; 4College of Materials Science and Engineering, Northeast Forestry University, Harbin 150040, China; 5College of Foreign Languages, Northeast Forestry University, Harbin 150040, China

**Keywords:** polyethylene terephthalate, beverage packaging, stress cracking, materials, structure, process flow, optimization, simulation, review

## Abstract

Compared with other materials, polyethylene terephthalate (PET) has high transparency, excellent physical and mechanical properties in a wide temperature range and good hygiene and safety, so it is widely used in the packaging industry, especially in the packaging of beverages and foods. The optimization of PET bottles is mainly reflected in three aspects: material optimization, structure optimization and process optimization, among which there is much research on material optimization and process optimization, but there is no complete overview on structure optimization. A summary of structural optimization is necessary. Aiming at structural optimization, the finite element method is a useful supplement to the beverage packaging industry. By combining the computer-aided design technology and using finite element software for finite element simulation, researchers can replace the experimental test in the pre-research design stage, predict the effect and save cost. This review summarizes the development of PET bottles for beverage packaging, summarizes various optimization methods for preventing stress cracking in beverage packaging, and especially focuses on comparing and evaluating the effects of several optimization methods for packaging structure. Finally, the future development of all kinds of optimization based on structural optimization in the field of beverage packaging is comprehensively discussed, including personalized design, the combination of various methods and the introduction of actual impact factor calculation.

## 1. Introduction

In 1909, Leo Baekeland invented the first synthetic plastic: phenolic plastic. The appearance of phenolic resin laid the foundation for the invention and production of various plastics and heralded the arrival of the plastic era [1,2,3,4,5,6]. The plastic manufacturing industry developed rapidly during the Second World War, benefiting from the progress of the petrochemical industry [7,8,9,10,11,12]. Polyethylene terephthalate (PET) was invented in the 1940s. It has the characteristics of light weight, high strength, good dimensional stability and no toxicity to the human body [13,14,15]. In the 1970s, Coca-Cola Company introduced PET bottles to the market for the first time and soon gained a leading position in the packaging field, not only replacing glass but also replacing metal cans to some extent. PET is widely used in many fields, especially in the packaging of bottled drinks [16,17,18,19,20,21]. In 2018, PET accounted for almost 16% of plastic consumption in the European packaging industry. So far, it has become one of the most used beverage packagings in the world, and it is still increasing [22].

Due to the limitation of chemical technology and mechanical technology, the performance of PET produced in early mass production is far from today’s level. Therefore, the bottom is designed to be hemispherical to disperse the pressure to the greatest extent, and is equipped with a detachable base part to keep the bottle upright [23]. However, with the progress of science and technology, the PET bottle with hemispherical bottom has been replaced by more personalized and diversified shapes, such as the concave pentagonal petal bottom, the bottle bottom with radial grooves and the claw-petal bottom structure represented by Coca-Cola Company [24,25,26,27]. These one-step blow molding designs reduce the production cost, simplify the production steps, and are more convenient and environmentally friendly. However, compared with the hemispherical base, it is easier to crack, which is caused by the mechanical properties given by the microscopic crystallinity and molecular orientation [28,29,30,31,32,33], and the more obvious stress concentration in the macroscopic view [34,35,36,37,38].

In order to prevent cracking, researchers optimized the bottle, which was divided into three directions: material optimization, process optimization and geometric structure optimization (Figure 1). It is worth noting that researchers seem to be more concerned about the consequences of the operation in the blow molding process than the influence of the structural design of the bottle itself on the final product. Therefore, although the finite element analysis of the PET beverage bottle itself is an effective way to enhance its mechanical properties, which is different from industrial preparation and chemical synthesis, there are few related research works and no summary of research progress. For this reason, this paper will include and pay attention to the progress of structure optimization of PET beverage bottles.

This paper summarizes the development of PET bottles for beverage packaging and the existing stress cracking problems (Part 1), enumerates the application of the finite element method to prevent stress cracking in beverage packaging, and compares and evaluates several packaging optimization methods, especially the structural optimization methods (Part 2). Finally, the application prospects of structural optimization in this field are summarized and discussed (Parts 3 and 4).

## 2. Material Optimization

Researchers usually add various chemicals to PET to give the bottles more functions [39,40,41], such as high temperature resistance [42,43], gas barrier [44,45,46] and sterilization [47] (Table 1). This does bring more opportunities to the packaging industry, but the increase in cost is a real problem. More importantly, a large amount of literature shows that the extensive use of PET will bring harm to the environment and human health [48,49,50,51,52,53,54,55,56,57]. In terms of the environment, Ajaj et al. [48] proposed that floatable plastics can affect water quality by increasing the risk of regional flooding. Thompson et al. [49] put forward that the use of plastics is unsustainable, considering the reduction of fossil fuel reserves and the limited ability to treat landfill waste. Kumartasli and Avinc [50] focused on the ocean and pointed out that plastics, as one of the most important components of marine debris, would cause great damage to natural habitats and ecosystems.In addition, it is worth noting that PET can leach out chemicals harmful to health, such as antimony trioxide, bisphenol A and phthalates [54,55,56,57]. Cooper and Harrison [55] found that the temperature in closed spaces (such as cars) in summer will promote antimony leaching from water, which will bring along physiological discomfort, reproductive damage and potential mutagenicity and carcinogenesis. Kehinde et al. [56] pointed out that bisphenol A precipitated from plastics will increase the risk of pain and metabolic disorder, and especially act on women’s health problems such as endometrial hyperplasia, recurrent abortion and infertility. Sax [57] pointed out that phthalates in PET have great influence on human reproductive system, especially on the growth and development stage of infants and children. Therefore, more factors still need to be considered for the improvement of materials, and there is a long way to go.

The mechanical properties of the product are considered by most researchers [68]. Demirel et al. [69] found that the addition of Mg_2_B_2_O_5_ to the PET matrix significantly reduced the degradation of PET to acetaldehyde, carboxylic acid and diethylene glycol, and increased the intrinsic viscosity of the composites. Inaner et al. [58] prevented the photocatalytic degradation of the PET bottle by adding CaB_2_O_4_into the blend and improved the mechanical properties of the PET bottle, that is, the bearing capacity was about 109% higher than that of pure PET. Inaner’s research results ensure that food is protected from the harmful effects of light, and can prevent the deformation of PET packaging materials due to various reasons. Similarly, Kocayavuz et al. [59] doped Ca_3_B_2_O_6_ synthesized by sol-gel method into PET, which significantly improved the mechanical properties of PET and increased the environmental stress cracking time from 0.3 min to 18 min. In addition, the ultraviolet transmittance of PET in the visible region decreased to ~18%. This method effectively solves the problems of packaging cracking and photocatalytic degradation of PET food and beverage packaging during storage. In addition, graphite nanosheets, montmorillonite, nano-hydroxyapatite and AlOx are also common mixed materials, which can improve the ultraviolet-visible light barrier property and oxygen barrier property [60,63,70,71]. Antibacterial PET is a new type of functionalization proposed in response to the requirements of food and beverage products [72,73,74,75,76]. 

With the progress of science and technology, it is the research trend to integrate various functions into one product. Mousavi et al. [77] found that the addition of acryl butadiene styrene (ABS) to PET can lead to an increase in tensile strength, while it can lead to a decrease in elongation at break and Young’s modulus. In addition, the addition of oak husk and potassium sorbate to the PET/ABS blend enhanced the antimicrobial properties. It can also significantly improve the water absorption and oxygen permeability in the PET/ABS mixture. In a recent study, Ahmedet al. [78] used 2,2’-Bifuran-5,5’-dicarboxylic acid (BFDCA) to modify PET, and got a more versatile new PET with a slightly higher glass transition range, higher tensile modulus, enhanced oxygen barrier property and excellent ultraviolet barrier property, while maintaining good transparency, which is ideal for advanced packaging applications.

## 3. Process Optimization

Environmental factors in the production process have great influence on the final molding, among which the biggest influencing factors are temperature and air pressure [79,80,81]. Although the study of environmental factors has taken practicality into consideration and provided guidance for industrial production to a certain extent, due to the fact that different machine models and stretching methods in real life have an influence on the presentation of the final sample, more comprehensive studies of factors are needed [82,83,84,85].

### 3.1. Temperature

McEvoy et al. [86] considered the influence of temperature on material creep, analyzed the creep constitutive model by ABAQUS finite element method, and predicted the thickness, strain and blow molding pressure of the bottle sidewall according to the temperature change between 90–110 °C in the production process. The results obtained were consistent with those obtained by commercial process conditions. Cosson et al. [87], who also consider the creep of materials at temperature, analyzed the high viscosity and strain hardening effect of materials at temperature range t > t_g_, and determined a simple viscoplastic model according to the results of uniaxial and biaxial tensile tests. On the relationship between thickness and initial temperature distribution, Hong et al. [88] thought that the initial temperature distribution of bottle blank is the most important factor affecting the precision of container blow molding. Therefore, all the heat transfer processes in the injection-blow molding process are considered by finite element analysis, and the factors affecting the thickness of finished products are predicted. Michels et al. [89] developed a comprehensive simulation concept to describe the elastic modulus as a function of process parameters such as local stretching and die temperature, and developed a new algorithm to identify the degree and direction of local stretching by using the results of process simulation.

### 3.2. Air Pressure

Kim and Seol [90] focused on the study of the influence of air pressure on the thickness of bottle wall during blow molding, and carried out finite element simulation with ANSYS Polyflow to measure the influence of air pressure on the wall thickness. In fact, if the preform parameters and all environmental factors can be comprehensively considered, more accurate production process parameters can be determined in industrial manufacturing. Tan et al. [91] developed a two-dimensional isothermal finite element simulation of the ISBM process for PET containers by commercial finite element software ABAQUS/Standard. The results show that the simulated constant mass flow method with constant mass flow rate as input is more suitable for simulating the blow molding stage in the ISBM process. Bagherzadeh et al. [92] used the finite element method to numerically model stretch blow molding (SBM) of the PET bottle and applied the superelastic constitutive model to different high temperatures and strain rates, and obtained the relationship between heat transfer coefficient, initial pre-blown air inlet time delay and bottle thickness.

### 3.3. Others

The thickness of the side wall is the main parameter of the bottle body, and it is also the focus of researchers’ attention in the production process. Atigkaphan and Thusneyapan [93] comprehensively considered temperature and air pressure, applied finite element analysis to the simulation of the production process, and predicted the thickness of four sides of a square cross-section bottle. Compared with the experiment, the predicted thickness deviation was within 3.573%. Chung [94] used ABAQUS to simulate the stretch blow molding process of the PET bottle by the finite element method. According to the movement of the plunger and the change of gas pressure, it was found that using a plunger significantly contributed to the uniform distribution of bottle sidewall thickness. Haddad et al. [95] studied the finite element simulation of the ISBM process through B-SIM simulation software and thought that a PET bottle produced by injection blow molding process could produce more uniform thickness distribution, improve the quality of PET bottle and shorten the production time.

## 4. Structural Optimization

The structural parameters of preform have great influence on the final product [96,97]. It is worth mentioning that most of these optimization analyses use the symmetry hypothesis, that is, divide the regular 360° cross section of the bottle in the vertical direction into five equal parts of 72° (or equal parts of other angles), and only analyze a certain small part, which is applicable in almost all PET process or structure improvement studies, and saves calculation resources and reduces calculation time. Figure 2 shows the application of the symmetry hypothesis.

### 4.1. Single Optimization

Lontos and Gregoriou [98] considered three different preform lengths in order to explore the influence of preform length on the wall thickness distribution of the final product and found that the bottom area of the bottle made of the longer preform length was thicker, which enhanced the overall stability. In order to minimize the wall thickness and reduce the use of PET, Tan et al. [99] used ANSYS Polyflow to evaluate the wall thickness distribution of PET bottles with different diameters of initial bottle blanks and determined that the allowable wall thickness of preforms with at least 6 mm diameter can be 1.3 mm. Sidorov et al. [100] established the relationship between the diameter, wall thickness and the length of forming zone of billet and product, and believed that the most acceptable diameter of billet should be within 30% of the diameter of the finished product.

In fact, in the optimization of PET bottle structure, compared with the bottle body with uniform structure, the bottle bottom has become the most studied object because of its diverse shapes and complex structures. Another reason is that the liquid leakage caused by rupture mostly occurs at the bottom of the bottle. The following will list and analyze different ways of bottle bottom structure optimization.

#### 4.1.1. Single Manual Optimization

Manual adjustment of the inherent parameters one by one is the earliest method used [101]. Common adjustable parameters include bottle bottom inner diameter, valley bottom inclination, valley bottom deepest, outer diameter and claw number (Figure 3). Although this method is effective, it still has the following obvious disadvantages:

① It takes energy to adjust one by one, and the manually set values must be discontinuous. The values between the two adjustments have not been applied and discussed.

② Because it is impossible to directly compare the influence of one parameter change on other parameters, conflicts among variables may arise.

Song et al. [102] found that the maximum principal stress decreased with the increase of the opening angle of the groove side wall, the arc diameter of the groove bottom and the number of claw petals in the claw-petal PET bottle bottom through the single factor test. It is indirectly confirmed by an all-factor experiment that the diameter of the circular arc at the bottom of the groove, the number of grooves and the depth of grooves have a significant influence on the maximum principal stress at the bottom of the bottle. Lyu et al. [103] have thought that the cracking phenomenon of the PET bottle is not only caused by insufficient tensile strength but also caused by rough design. However, at first, they only explored the change of the maximum principal stress of bottles with three different thicknesses, and there was no clear structural optimization. Later, they conducted further research on the structural design [104,105] and manually adjusted the three design factors: clearance, length and valley width, which were found to dominate the cracking of the bottle bottom in the cracking test, so that the maximum principal stress decreased by 21%. Yuan et al. [106] used ANSYS software to analyze the strain of PET beer bottles by finite element method under two loads of different thicknesses and bottle diameters and explored the influence of thickness and bottle diameter on the mechanical properties of PET beer bottles. Demirel and Daver [107,108] used numerical simulation and finite element analysis techniques to redesign the foot length, valley width and gap of the PET bottle bottom, which reduced the maximum principal stress of the optimized bottle bottom surface by 10.8% and improved the stress cracking resistance by 88% in the accelerated stress cracking test.

#### 4.1.2. Single Automatic Optimization

With the development of computer technology, in order to realize the synergistic effect among parameters, automatic optimization that can get rid of changing data values one by one came into being [109]. Please note that the automatic optimization here refers to the mixed calculation of multiple factors, rather than the operation of continuous parameter changes of the same variable. Compared with manual optimization, the accuracy of automatic shape optimization has been greatly improved [110]. After a certain variable undergoes a slight change, several groups of functions will immediately calculate the change of the overall stress, and guide other variables to adjust through function or exhaustive method, forming a feedback mechanism. In fact, there are some disadvantages:

① Multiple optimization iterations will consume a lot of computing resources, with a large amount of computation and a long computation time.

② In order to ensure that the overall shape of the bottle is unchanged and easy to place and hold, shape optimization is required to be fine-tuned only in a small range. For macroscopic parameters, especially the inclination of the valley bottom, it is difficult to complete the overall change through automatic optimization.

When Huang et al. [111] designed the milk bottle, they discussed the influence of the thickness of the bottle body and the thickness of the bottom on the critical load and weight of the HDPE bottle. The weight of the redesigned bottle was reduced by 21.4% under the same top load. It should be pointed out that, although only thickness changes are involved, the thickness divided into several independent parts (bottle body, bottle bottom, bottle handle, joint, etc.) can be regarded as the mixed calculation of multiple factors. Comsol is a common optimization software, and its general optimization steps are shown in Figure 4. In the latest research [112], through numerical simulation and finite element analysis, we first analyzed four typical bottle bottom models and determined a better structure. Then, the automatic optimization method is used for fine-tuning. On the premise of the same material quality, the surface maximum principal stress, the overall maximum principal stress and the total elastic strain energy of the bottle bottom are reduced by 46.39–71.81%, 38.16–71.50% and 38.56–61.38%, respectively, and the deformation displacement is also reduced by 0.63–3.43 mm.

### 4.2. Manual-Automatic Double Optimization

In order to solve the shortcomings of the above single manual/automatic optimization, Ge-Zhang et al. [101] used a brand-new manual-automatic double optimization, that is, first manually control variables for preliminary comparison and optimization, and then automatically optimize on the premise of keeping the quality of consumable materials unchanged. The total maximum principal stress and total elastic strain energy of the bottle bottom decreased by 69.4% and 40.0%, respectively, and the displacement caused by deformation decreased by 74.1%. The proposal of manual-automatic double optimization not only inherits the high precision of automatic optimization but also saves computing resources by manually preprocessing macro-adjustment parameters, thus avoiding the disadvantages of too long a time and only micro-adjustment of automatic optimization. It can be seen from the alarming decline that this is a promising new method.

It can be seen from the comparison in Table 2 that the reduction ratio of maximum principal stress of manual-automatic double optimization and single automatic optimization is much better than that of single manual optimization. The number of iterations indicates that manual-automatic double optimization can achieve convergence with less computation, which also means that optimizing the same bottle will cost less time [101]. In fact, we think that when the goal is to design a new type of bottle, the advantages of manual-automatic double optimization can be more reflected: manually testing and optimizing the macro appearance, and automatically optimizing the details.

## 5. Summary

This review summarizes the development of PET bottles for beverage packaging and compares and evaluates the effects of several packaging optimization methods. Finally, the development prospect and challenges of PET optimization in the field of beverage bottle packaging are discussed. The optimization methods of the PET bottle summarized in this paper are mainly in three aspects: material optimization, process optimization and structure optimization. Among them, material optimization can improve the mechanical properties of PET bottles from the characteristics of the bottles themselves, while the introduction of substances enhances the multifunctional development of the bottles, such as enhancing the oxygen resistance and antibacterial properties. In terms of process flow, this paper mainly summarizes the optimization of PET bottle performance from the aspects of temperature, air pressure, etc. Generally speaking, changing the initial temperature distribution of bottle blank is the most important factor affecting the container blow molding accuracy. As the key part of structural optimization, the manual optimization method based on finite element is an early optimization method with low accuracy. The automatic optimization method based on finite element is the mainstream nowadays, which has high accuracy but takes a long time. The manual-automatic double optimization method based on finite element combines the advantages of the two methods and achieves high accuracy on the premise of moderate calculation, which may become the frontier method of finite element optimization of the bottle model (Figure 2). However, it is undeniable that no matter what structural optimization method is adopted, structural optimization cannot replace the influence of material composition and environment on the cracking degree of finished products. To make the best bottle, it is necessary to combine material selection, bottle design, engineering optimization and other fields, which also means greater workload.

## 6. Outlook

The optimization design of PET beverage bottles can be divided into three aspects: material optimization, process optimization and physical structure optimization, among which the research on material optimization and process optimization is rich. Chemical-based material optimization provides versatility, which is in line with the development trend of new packaging. Process optimization shows and improves the process from laboratory to factory; Physical-based structural optimization is mainly aimed at the improvement of mechanical properties. When we design a new bottle or improve the old bottle, the best way is to integrate the three optimizations, but we have to admit that the engineering quantity is also increasing. For the future optimization design, the main challenges are as follows:

(1) Multifunctional or even all-round beverage bottle design is the future trend, which can be accomplished through material optimization. However, as mentioned before, it is important to pay attention to the cost of the material (the price of silver ions used for antibacterial) and whether it is harmless to the human body (requiring long medical research). Similarly, the feed inlet of the production machine may also need to be improved.

(2) The current process optimization is based on the existing process improvement, and many actual production devices can’t meet the rigorous design requirements assumed in the research. The process flow is applied to reality, so the feasibility of the design should be considered. Perhaps it is easier for manufacturers to accept the parameters in a general range or improve the machinery.

(3) Although structural optimization can reduce the maximum principal stress and stress deformation of the bottle, the change of physical shape obviously can’t replace the multi-functional trend brought by material optimization. The above two methods should be combined to change the parameters of new materials according to the materials used, and whether the actual production equipment can meet the requirements of the model should be considered.

## Figures and Tables

**Figure 1 polymers-14-03364-f001:**
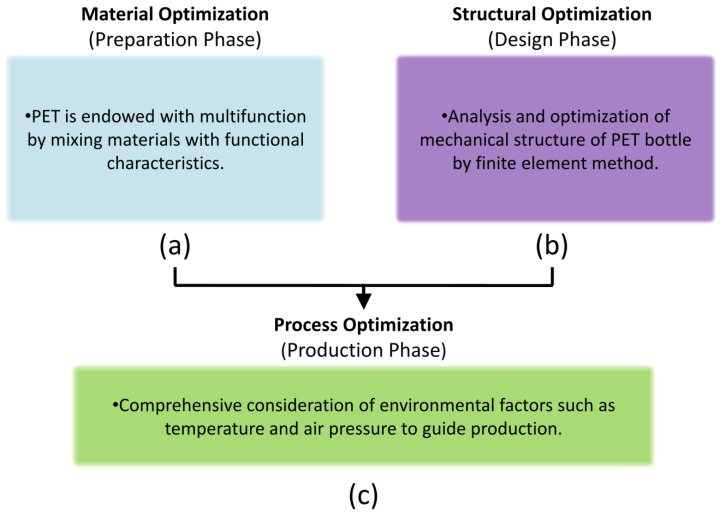
Three optimization methods of PET bottle.

**Figure 2 polymers-14-03364-f002:**
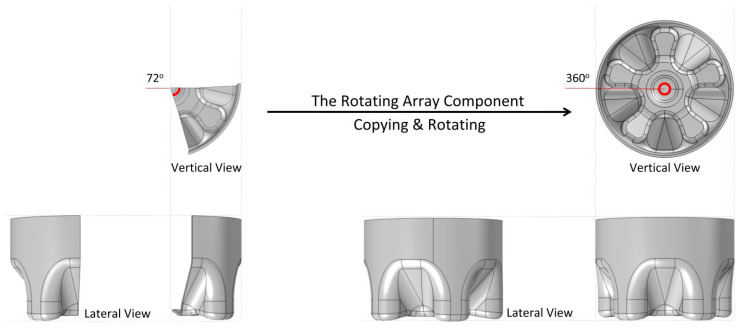
Symmetry hypothesis for fast modeling and calculation.

**Figure 3 polymers-14-03364-f003:**
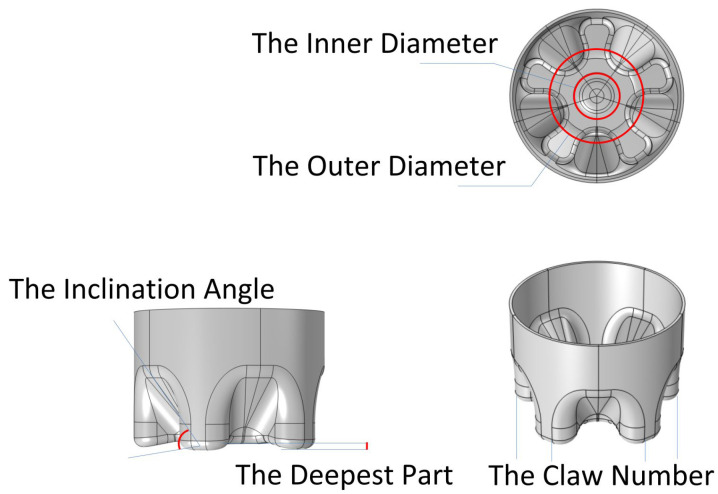
Common adjustable parameters of bottle bottom.

**Figure 4 polymers-14-03364-f004:**
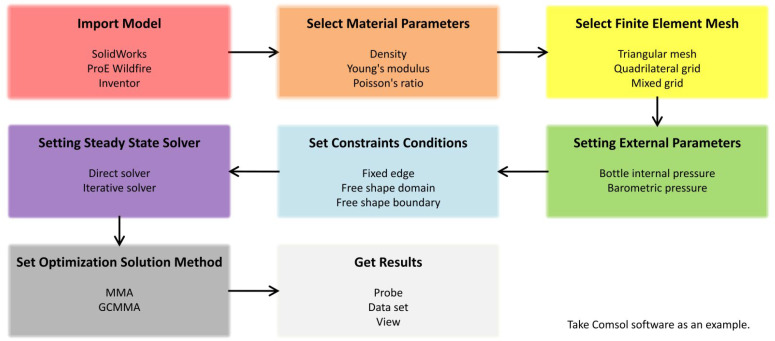
General steps of optimizing the structure of PET bottles.

**Table 1 polymers-14-03364-t001:** Multifunctional PET by adding raw materials.

Characteristics	Blend	Test Index	Effect	Ref.
Light Blocking	CaB_2_O_4_	UV transmittance	Decrease by ~88%	[58]
	Ca_3_B_2_O_6_	UV transmittance	Decrease by ~67%	[59]
	nHAp	Visible light transmission	Decrease by ~80%	[60]
High Temperature Resistance	Montmorillonite (MMT)/Laponite (LAP)/Polyvinyl alcohol (PVA)	Exothermic rate Total exothermic peak	Decrease by 67.4% Increase by 45.3%	[61]
	Gelatin-basedCarbon Dots	Peak value of heat release rate Total smoke production	Decrease by 42.66% Decrease by 62.64%	[62]
Gas Barrier	Graphite nanoplatelets (GNPs)	Oxygen transmission rate	Decrease by >99%	[63]
	Terephthalate-intercalated LDHs	Oxygen transmission rate	Decrease by 46.2%	[64]
	NK75 nanoclay	Oxygen transmission rate	Decrease by 38%	[46]
	Nanotalc nanohybrids	Oxygen transmission rate	Decrease by 64%	[65]
	Polyelectrolyte/Clay Coacervate	Oxygen transmission rate	Decrease by more than three orders of magnitude	[66]
Sterilization	Phosphorylated chitosan/Al nanoparticles	The number of Escherichia coli	Decrease by ~2/3	[67]
Mechanical Property	CaB_2_O_4_	Carrying capacity	Increase by ~109%	[58]
	Ca_3_B_2_O_6_	Carrying capacity	Increase by ~133.66%	[59]
	Rubber	Toughness	Increase by 85%	[68]
	Terephthalate-intercalated LDHs	Tensile strength Young’s modulus	Increase by 29.4% Increase by 38.9%	[64]
	NK75 nanoclay	Young’s modulus	Increase by 66%	[46]

**Table 2 polymers-14-03364-t002:** Comparison of optimized data of claw-petal PET bottle.

Structural Optimization Method	Surface Maximum Principal Stress Reduction Ratio	Iterations	Ref.
Single manual optimization	8.10–~52%		[101,102,105,107]
Single automatic optimization	66.90%	20	[101,112]
Manual-automatic double optimization	67.86%	17	[101]

## Data Availability

Not applicable.

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
