# Peer review of "Advances in Polyethylene Terephthalate Beverage Bottle Optimization: A Mini Review"

_polymers, 2022, doi:10.3390/polym14163364_

Round 1

Reviewer 1 Report

Comment for polymers-1867257 is listed as follows,

1. There are some miss been named or error typing in the pdf file of manuscripts.

(1) In the Keywords: please change the "Simulations" into the "Simulation".

(2) In the subsection 3.1. Temperature: please change the "Cossonet et al. [77]" into the "Cosson et al. [77]".

(3) In the subsection 3.2. Air Pressure: please change the "Kim et al. [80]" into the " Kim and Seol [80]".

(4) In the subsection 3.3. Others: please change the "Atigkaphan et al. [83]" into the " Atigkaphan and Thusneyapan [83]". Also please change the "Chung et al. [84]" into the "Chung [84] ", change the "Hadad et al. [85]" into the "Haddad et al. [85]".

(5) In the subsection 4.1. Single Optimization: please change the "Lontos et al. [88]" into the "Lontos and Gregoriou [88]".

(6) In the subsection 4.1.1. Single Manual Optimization: please change the "Demirel et al. [97,98]" into the "Demirel and Daver [97,98]".

(7) In the subsection 4.2. Manual-Automatic Double Optimization: please change the "Ge et al. [91]" into the "Ge-Zhang et al. [91]".

(8) In the Table 1: please check "- ~ " of the "UV transmittance - ~88%", check "- >" of the "oxygen transmission rate - > 99%", check "+" of the "Toughness + 85%", etc.

(9) In the Table 2: please check "- ~ " of the "8.10% - ~52%".

Reviewer 2 Report

Yes, PET plastic is BPA free, but it is well known that PET  leaches antimony trioxide and phthalates which are dangerous to health. Antimony may contribute to cancer development, skin problems, menstrual and pregnancy issues, phthalates are endocrine disruptors. This problem has not been commented in review in detail though food safety is very important in PET application in food packaging. At present at the end of Introduction the struture of manuscript is briefly described, It would beneficial to clearly and more specify the objective of the review.

The big environmental problem that PET is not biodegradable and contains harmful substances. The statement in lines 78-80  needs citation and more comments because a lot of publications report harmful substances in PET. 

This review provides useful information regarding material optimization, process optimization and structure optimization.

Round 2

Reviewer 1 Report

Comment for polymers-1867257-v2 is listed as follows,

1. There is one miss been named in the pdf file of manuscripts.

(1) In the section 2. Material Optimization: please change the "Kumartasli et al. [50]" into the "Kumartasli and Avinc [50]".

Author Response

Thank you for your careful review and quick reply! We have revised the error you pointed out in the article.

Page 3, Line 84:
"Kumartasli et al. [50]" → "Kumartasli and Avinc [50]".
